# Acoustic Metamaterials for Low-Frequency Noise Reduction Based on Parallel Connection of Multiple Spiral Chambers

**DOI:** 10.3390/ma15113882

**Published:** 2022-05-29

**Authors:** Haiqin Duan, Fei Yang, Xinmin Shen, Qin Yin, Enshuai Wang, Xiaonan Zhang, Xiaocui Yang, Cheng Shen, Wenqiang Peng

**Affiliations:** 1College of Field Engineering, Army Engineering University of PLA, Nanjing 210007, China; dhq1135168523@163.com (H.D.); 19962061916@163.com (F.Y.); wangenshuai0823@126.com (E.W.); zxn8206@163.com (X.Z.); 2Engineering Training Center, Nanjing Vocational University of Industry Technology, Nanjing 210023, China; 2019101052@niit.edu.cn; 3MIIT Key Laboratory of Multifunctional Lightweight Materials and Structures (MLMS), Nanjing University of Aeronautics and Astronautics, Nanjing 210016, China; cshen@nuaa.edu.cn; 4College of Aerospace Science and Engineering, National University of Defense Technology, Changsha 410073, China; plxhaz@126.com

**Keywords:** acoustic metamaterial, noise control, multiple spiral chambers, Fabry–Perot resonance, low-frequency sound absorption, finite element simulation

## Abstract

Acoustic metamaterials based on Helmholtz resonance have perfect sound absorption characteristics with the subwavelength size, but the absorption bandwidth is narrow, which limits the practical applications for noise control with broadband. On the basis of the Fabry–Perot resonance principle, a novel sound absorber of the acoustic metamaterial by parallel connection of the multiple spiral chambers (abbreviated as MSC-AM) is proposed and investigated in this research. Through the theoretical modeling, finite element simulation, sample preparation and experimental validation, the effectiveness and practicability of the MSC-AM are verified. Actual sound absorption coefficients of the MSC-AM in the frequency range of 360–680 Hz (with the bandwidth Δ*f*_1_ = 320 Hz) are larger than 0.8, which exhibit the extraordinarily low-frequency sound absorption performance. Moreover, actual sound absorption coefficients are above 0.5 in the 350–1600 Hz range (with a bandwidth Δ*f*_2_ = 1250 Hz), which achieve broadband sound absorption in the low–middle frequency range. According to various actual demands, the structural parameters can be adjusted flexibly to realize the customization of sound absorption bandwidth, which provides a novel way to design and improve acoustic metamaterials to reduce the noise with various frequency bands and has promising prospects of application in low-frequency sound absorption.

## 1. Introduction

The study of the impact of noise on environmental pollution is very important to environmental researchers and specialists, because it has a significant impact on human life [1]. The most intuitive influences of noise to human beings include annoyance, sleep disturbance, and interference with communication [2,3]. Noise also interferes with complex task performance, modifies social behavior, and causes many physical and psychological problems when the time of exposure to noise is long [4]. The major health problems that can result from noise pollution include: damage to the hearing that, in turn, may lead to depression, anxiety, social isolation; intensification of health problems, irritation, stress, temporary increase in blood pressure, weakened immune system, heart disease, and cardio-circulatory problems [5]. Moreover, new buildings with small spaces and thin walls require new absorbing technologies to obtain effective noise reduction with limited structural size [6,7,8]. The sound absorption performances of a spent coffee grounds/potato starch bio-based composite were analyzed by Moussa et al. [6] for potential application in buildings, which aimed to explore new alternative and low-environmental-impact materials, with a particular focus on bio-sourced materials. Furthermore, the frequency region of a varied noise source is different most of time, and each sound-absorbing material or structure has a limited effective absorption band normally, which indicates that design of the sound absorber must take the limitations to the selected frequency bandwidth of the noise source into consideration [9,10,11]. Therefore, it is urgent to develop novel sound-absorption materials and structures to control the environmental noise for improvements in living conditions and quality of life.

In recent years, acoustic metamaterials (AMs) with subwavelength sizes have attracted much attention, such as the membrane AM [12,13,14,15,16], the Helmholtz AM [17,18,19] and the Fabry–Perot (FP) AM [20,21,22,23], which can gain excellent sound absorption characteristics in the deep-subwavelength noise frequency range. The membrane AM is usually composed of an elastic membrane and rigid disk. Due to the small restoring force of the elastic membrane, it is easy to achieve resonance in the audible frequency range. Yang et al. [12] first reported that a circular membrane AM with a diameter of 20 mm and a thickness of 0.28 mm resonated at 146 Hz to achieve perfect sound absorption. The Helmholtz resonance AM is a classic sound-absorbing structure, which can be utilized to enhance the absorption of sound energy and precise control of the sound absorption frequency can be achieved through the adjustment of structural parameters [18]. Acoustic absorption can also be achieved through FP resonance. Cai et al. [20] reported such an absorber, which coiled a 205 mm FP channel into a space with a thickness of 9.7 mm; near perfect absorption performance was observed at about 400 Hz in the experimental validation. AMs with the deep-subwavelength size can achieve significant sound absorption at specific frequencies, but most of these AMs just obtain a narrow sound absorption bandwidth and can only absorb noise with several specific frequencies, which is limited by the resonant nature [24]. It cannot meet the actual requirements of broadband noise reduction, which limits practical applications in the field of low-frequency noise control. Therefore, it is a challenging task to develop small-sized structures to control the low-frequency noise.

The most effective strategy to broaden the sound absorption bandwidth is to combine the various resonant response units [25,26,27,28,29,30,31,32,33,34,35,36,37]. It is difficult to accurately modulate the resonance frequencies of multiple units in the membrane AM due to the difficulty in controlling the membrane tension, so it is seldom used in low-frequency broadband noise control. Coupling of multiple Helmholtz resonances to achieve broadband sound absorption is the most popular method used by researchers. Guo et al. [25] designed an acoustic metamaterial composed of 16 Helmholtz resonators, which achieved broadband sound absorption in a frequency range of 700–1000 Hz through parameter optimization. The structural thickness was 1/25 of the noise frequency wavelength. Liu et al. [26] proposed a multilevel Helmholtz metamaterial with deep subwavelength thickness, composed of eight Helmholtz resonators, which can achieve perfect continuous acoustic absorption in a range of 400 Hz to 2800 Hz. The topology optimization was treated as a tool by Sharma et al. [27,28] to design the acoustic metamaterials with the wideband width, and the reported numerical framework and inferences could show their potential use in the optimal design of soft compressible composites utilized in acoustic applications. All the above studies realized broadband sound absorption in a certain frequency range, but the defect of Helmholtz metamaterial is that the resonance frequency is determined by the structural parameters. In order to achieve the coupling of different sound absorption frequencies, the different chamber combinations with different structural parameters are required, which results in the difficulty of compact arrangement in the combined structures and a waste of space. The FP chamber can realize the phase modulation of the reflected sound wave by adjusting the length of chamber, thereby realizing effective manual control of the resonance frequency. The advantage of an FP chamber is that it can make sound waves propagate along a coiled path through crimping, folding, and other chamber forms. Researchers have carried out related research on the FP AM [29,30,31,32,33,34,35,36]. By using the coiled-up method, Zhang et al. [29] proposed an absorber based on six coiled FP channels, which achieved perfect absorption at 100–200 Hz and the requirement of deep-subwavelength thickness (0.07 *λ*) can still be satisfied. Based on the theory of double non-uniform cross-section channels with different widths, a double non-uniform cross section proposed by Han et al. [31] was applied to AM with the 16 FP channels, which effectively reduced the lower cut-off frequency in the sound-absorbing structure. The flexible designs greatly reduce the 1/4 wavelength *λ* scale required for FP channel resonance by using a crimped form and combining FP channels with different resonant frequencies to achieve wide sound absorption, with accurate design and layout. The combination of FP chambers with various length can achieve broadband sound absorption. However, the overall size of structure depends on the longest chamber, resulting in a waste of space. By folding the long chamber to the end of short chamber, the structure size is reduced to a certain extent. However, the fold is detrimental to sound absorption, because the sound spread in the folded chamber is disturbed.

On the other hand, to achieve the perfect sound absorption, the acoustic impedance of the FP channel is matched to the acoustic impedance of the air, which means that the length of the channel $L_{N}$ should be close to the 1/4 wavelength [37]. In the low-frequency region, the thickness of a quarter wavelength represents a very large structure (λ=c/f). To overcome this shortcoming, the effective propagation path (phase delay) is enlarged by coiling the path into a spiral to reduce the space [38,39,40]. The spiral chamber with gradient length is formed through the ingenious structure form, and the coupling effect from different chambers realizes the broadband perfect sound absorption. In this study, based on FP resonance theory, the novel FP channel with spiral mode is proposed, which is named as multiple spiral chambers acoustic metamaterial (MSC-AM). The FP channel is designed as a spiral chamber, which can compact the chambers with various lengths and reduce the occupied space of the MSC-AM structure. It overcomes the defect that the FP resonance requires the structure size to reach the 1/4 wavelength at the resonance frequency and not be suitable for low-frequency (which corresponds to the long wavelength) noise absorption. At the same time, the MSC-AM is composed of multiple spiral FP chambers with the same cross section, without other complex internal structure, and can be fabricated at one time by additive manufacturing. The sound absorption mechanism of the MSC-AM is investigated by the theoretical modeling and finite element simulation, and its effectiveness is validated by experimental detection. Broadband sound absorption is realized through the coupling of spiral chambers with different lengths in the MSC-AM. Moreover, the precise design of parameters can realize the customization of the sound absorption spectrum, which is conducive to promote the practical application of MSC-AM.

## 2. Materials and Design

### 2.1. Structural Design

Schematic diagram of the MSC-AM structure is shown in Figure 1 and the spiral chambers with the same color in Figure 1a represent a group of spiral chambers with the same length. The 3D modeling software is used to design the structure. A spiral line is drawn as the guide line, and a 2D sketch of the chamber section is drawn. The 3D structure is constructed through scanning the sketch along the spiral line. In order to facilitate the arrangement and make full use of the space, the cross-section shape of the spiral chamber is square (or shape of periodic array, such as diamond or regular hexagon). In Figure 1c, the marked number represents a group of chambers with the same parameters, such as the number 1 represents the four blue chambers in Figure 1a; number 3 represents the four green chambers in Figure 1a; number 4 represents the eight yellow chambers in Figure 1a; number 6 represents the eight red chambers in Figure 1a.

The whole structure is a cylinder, and the spiral chamber is tightly embedded around the same spiral center. Each single spiral chamber has the same square cross-sectional shape with a side length *a*. As shown in Figure 1b,c, height of the structure is *H* (according to the actual needs, height of the structure can be greater or less than the pitch *p_N_*), and the spiral radius is *R_N_*, where *N* represents the *N*th group of chambers with the same parameters. The bottom of the chamber is closed, and the chambers are not connected to each other. The chambers near the center have a smaller length and those far from the center have a larger length, thereby forming the multiple sets of spiral chamber units with gradient lengths. Through selecting and adjusting the structural parameters, broadband reduction in the low-frequency noise can be achieved. Changing the pitch can adjust the length interval among the different chambers, which will result in a change in the position interval of different sound-absorption peak frequencies. A spiral chamber with a different length can independently modulate the incident sound wave. The delay phase ϕ of the spiral chamber and the chamber length $L_{N}$ satisfy the expression $\phi =k_{cN}L_{N}$. Here $k_{cN}$ is the effective transfer function, and the specific calculation formula is given in the theoretical modeling process. The phase of sound waves can be adjusted by adjusting the length of the spiral chamber.

### 2.2. Theoretical Modeling

Theoretical model of sound absorption coefficient α of the MSC-AM can be constructed according to its acoustic impedance [41,42,43], and the corresponding computational formula is shown in Equation (1). Here Z is total acoustic impedance of the MSC-AM; $\rho_{0}$ and $c_{0}$ are density and acoustic velocity of the air under normal temperature and normal atmospheric pressure, respectively.
(1)$$\alpha =1-\left|\frac{Z-\rho_{0}c_{0}}{Z+\rho_{0}c_{0}}\right|$$

Total acoustic impedance Z of the MSC-AM can be calculated by Equation (2), in which $Z_{N}$ and $M_{N}$ are the acoustic impedance and quantity of the chambers with the same parameters in the MSC-AM, respectively; M is total quantity of the chambers in the MSC-AM, and $M=\sum M_{N}$.
(2)$$Z=\frac{1}{\sum_{1}^{M}\left(1/Z_{M}\right)}=\frac{1}{\sum_{1}^{N}\left(M_{N}/Z_{N}\right)}$$

Acoustic impedance $Z_{N}$ of a single chamber in the MSC-AM can be derived by Equation (3). Here $Z_{tN}$ is the *N*th single chamber with the same parameters in the proposed MSC-AM, and $\sigma_{N}$ is the corresponding perforation ratio.
(3)$$Z_{N}=Z_{tN}/\sigma_{N}$$

The perforation ratio $\sigma_{N}$ can be obtained by Equation (4). Here $r_{N}$ is equivalent radius of the cross section of the spiral chamber; $a_{N}$ is length of side of the square section of the spiral chamber; A is the total area of the front panel of the MSC-AM.
(4)$$\sigma_{N}=M_{N}\pi r_{N}^{2}/A=M_{N}a_{N}^{2}/A$$

Total area A of the front panel of the MSC-AM can be achieved by Equation (5), in which R is the radius of cross section of the experiential sample.
(5)$$A=\pi R^{2}$$

Acoustic impedance $Z_{tN}$ of the single spiral chamber can be derived by Equation (6). Here $Z_{cN}$ is effective characteristic impedance of the air in the spiral chamber, which can be calculated by Equation (7); $k_{cN}$ is effective transfer function of the air in the spiral chamber, which can be obtained by Equation (8); $L_{N}$ is length of the spiral chamber, which can be achieved by Equation (9).
(6)$$Z_{tN}=-iZ_{cN}\cot\left(k_{cN}L_{N}\right)$$
(7)$$Z_{cN}={\left(\rho_{cN}/C_{cN}\right)}^{1/2}$$
(8)$$k_{cN}={\left(\rho_{cN}C_{cN}\right)}^{1/2}$$
(9)$$L_{N}=\sqrt{{\left(2\pi R_{N}\right)}^{2}+P_{N}^{2}}\cdot \frac{H}{P_{N}}$$

In Equation (9), $R_{N}$ is spiral radius of the single spiral chamber; $P_{N}$ is the corresponding pitch; H is total thickness of the MSC-AM. In Equations (7) and (8), $\rho_{cN}$ and $C_{cN}$ represent the effective density and the effective volume compressibility ratio of the air, respectively, and they can be calculated according to the thermal–viscous acoustic theory, as shown in Equations (10) and (11), respectively [41,42,43].
(10)$$\rho_{cN}=\rho_{0}\frac{va_{N}^{2}h_{N}^{2}}{4i\omega }{\left\{\sum_{m=0}^{\infty }\sum_{n=0}^{\infty }{\left[\alpha_{m}^{2}\beta_{n}^{2}\left(\alpha_{m}^{2}+\beta_{n}^{2}+\frac{i\omega }{v}\right)\right]}^{-1}\right\}}^{-1}$$
(11)$$C_{cN}=\frac{1}{P_{0}}\left\{1-\frac{4i\omega \left(\gamma -1\right)}{v^{\prime }a_{N}^{2}h_{N}^{2}}{\sum_{m=0}^{\infty }\sum_{n=0}^{\infty }\left[\alpha_{m}^{2}\beta_{n}^{2}\left(\alpha_{m}^{2}+\beta_{n}^{2}+\frac{i\omega \gamma }{v^{\prime }}\right)\right]}^{-1}\right\}$$

In Equations (10) and (11), v can be obtained by Equation (12); $h_{N}$ is also length of side of the square section of the spiral chamber, meaning $h_{N}=a_{N}$; ω is the acoustic angle frequency, which can be derived by Equation (13); $\alpha_{m}$ and $\beta_{n}$ are two intermediate constants, which can be calculated by Equations (14) and (15), respectively; $P_{0}$ is the standard atmospheric pressure under normal temperature; γ is the specific heat ratio of the air; $v^{\prime }$ can be achieved by Equation (16).
(12)$$v=\mu /\rho_{0}$$
(13)ω=2πf
(14)$$\alpha_{m}=\left(m+1/2\right)\pi /a_{N}$$
(15)$$\beta_{n}=\left(n+1/2\right)\pi /h_{N}$$
(16)$$v^{\prime }=\kappa /\left(\rho_{0}C_{v}\right)$$

In Equation (12), μ is the dynamic viscosity coefficient. In Equation (13), f is the studied acoustic frequency. In Equation (16), κ and $C_{v}$ are thermal conductivity and specific heat capacity at the constant volume, respectively. According to the above Equations (1)–(16), theoretical model for the sound absorption coefficient of MSC-AM can be constructed, and the utilized constants are summarized in Table 1.

Structural parameters of the proposed MSC-AM can be adjusted to achieve the variable sound absorption performance for the different requirements. In order to verify effectiveness and feasibility of the proposed MSC-AM, a group of structural parameters is selected for the given structure in Figure 1. Meanwhile, size for the detected sample required by the utilized standing wave tube detector is also taken into consideration. In this research, the MSC-AM in Figure 1 includes 6 groups of spiral chambers, which indicates N=6 and $M_{N}\left(N=1,2,3,4,5,6\right)$ are 4, 8, 4, 8, 8, and 8, respectively. Total quantity M of the chambers in the MSC-AM is 40. The cross section of each spiral chamber is the square section with the same parameters, which indicates that $a_{N}=h_{N}=10$ mm $\left(N=1,2,3,4,5,6\right)$. The spiral radius $R_{N}\left(N=1,2,3,4,5,6\right)$ of the 6 groups of chambers are 8.48 mm, 18.97 mm, 25.45 mm, 30.59 mm, 34.98 mm, and 42.42 mm, respectively, and all the pitches $P_{N}\left(N=1,2,3,4,5,6\right)$ are set as 100 mm. For the experimental sample, taking into account requirements of the standing wave tube measurement and the prospective application of the proposed MSC-AM, the radius R of the cross section is set as 50 mm and the total thickness H is set as 80 mm. As mentioned above, these structural parameters of the MSC-AM can be adjusted to realize the customization of the sound absorption spectrum for the various applications.

### 2.3. Theoretical Analysis and Experimental Validation

According to the constructed theoretical model of the MSC-AM in Section 2.2, the corresponding sound absorption performance and characteristics are theoretically analyzed. Afterwards, on the basis of the finite element simulation model, as shown in Figure 2a, the sound absorption mechanism of the MSC-AM structure is further revealed and investigated. The low force stereolithography (LFS) 3D printing method is used to prepare the experimental sample based on the additive manufacturing technique [44], and the utilized 3D printer is Form3 (supported by the Formlabs Inc., Summerville, MA, USA). The prepared experimental sample for the MSC-AM is shown in Figure 2b. Then the actual sound absorption coefficient of the prepared experimental sample is tested by standing wave tube tester AWA6290T (supported by Hangzhou Aihua Instruments Co., Ltd., Hangzhou, Zhejiang, China), as shown in Figure 2c, which can be treated as the experimental validation of sound absorption performance and bandwidth of the MSC-AM in this research [45,46].

#### 2.3.1. Finite Element Simulation Analysis

As a kind of artificial microstructure, the acoustic response of metamaterials mainly depends on the size, shape and arrangement of the micro-resonance units. Due to the complex composition of most metamaterials, there is no unified mathematical theory to predict the characteristic parameters of any kind of AM. Finite element analysis can simulate the real physical system with the mathematical approximation method and solve the complex problems by simplifying or decomposing them into simple problems. It is an effective analysis method for analysis of the characteristics for the AM, which can be adapted to the various complex shapes.

The main form of FP resonance acoustic energy dissipation is thermal viscosity dissipation in the resonant state. The thermal viscosity interaction module built into the simulation software is adopted to describe the air domain. As shown in Figure 2a, the background physical field (BPF) is a plane wave acoustic field with incident pressure amplitude of 1 Pa, which moves onto the surface of the structure along the negative side of the Z axis. The upper physical field is the perfectly matched layer (PML), which represents the infinitely distant air domain and absorbs all outgoing waves. A method of reducing mesh size and encrypting mesh division is used between the internal air domain and wall, and the viscous friction and thermal conduction in the structure are entirely considered. Considering that the stiffness of the MSC-AM structure is much higher than that of air medium, the boundary of spiral chamber adopts the hard boundary condition.

#### 2.3.2. Sample Preparation by Additive Manufacturing Technology

Excellent sound absorption performance of the AM absorber is achieved through ingenious design of its structure instead of the material property itself. In order to realize outstanding sound absorption characteristics for the AM, each structure unit should be accurately designed, which indicates that the complex structure in AM put forward extremely strict requirements for the preparation and fabrication technology. The novel additive manufacturing technique can improve manufacturing compliance to realize the integrated manufacturing of function and structure, which exhibits obvious advantages in the fabrication of complex structures for the AM. The LFS 3D printing method is utilized in this study to prepare the experimental sample of MSC-AM, which has the advantages of high forming precision and high material utilization ratio [44]. The Form3 3D printer is used to prepare the experimental sample for the MSC-AM in this research. The three-dimensional structure model of the designed MSC-AM is imported into the pre-processing software, and the processed data are transferred to the 3D printer after slicing. An ultraviolet light source with a wavelength of 405 nm and power of 250 mW is used to scan the photosensitive resin layer by layer, and the final experimental sample is formed through the layered accumulation. Size of the facula for the ultraviolet light is 85 μm, and the selected thickness of the single layer is 100 μm. During the additive manufacturing process, the temperature of the platform is controlled at 35 °C. After the preliminary fabrication, the sample is cleaned and further post-treated to improve its mechanical behavior.

#### 2.3.3. Experimental Verification by Standing Wave Tube

Standing wave tube method is the common technique to test the sound absorption coefficients of sound-absorbing materials or structures with vertical incidence [45,46]. The utilized AWA6290T standing wave tube tester is shown in Figure 2c. In the measurement, the sample to be tested is fixed in one side of the standing wave tube, and the plane wave with a certain frequency is generated in the other side through the loudspeaker. There will be absorption and reflection at the sample to be tested for the incident plane wave, and the standing wave is formed by superposition of the incident wave and the reflected wave in the standing wave tube. Acoustic pressure of the standing wave can be tested by an acoustic probe for different frequency automatically. The actual sound absorption coefficients $\alpha_{0}$ of MSC-AM with vertical incidence can be calculated by the peak value and valley value of the acoustic pressure [45,46], as shown in Equation (17).
(17)$$\alpha_{0}=\frac{I_{i}-I_{r}}{I_{i}}=1-\frac{I_{r}}{I_{i}}=1-\frac{P_{r}^{2}}{P_{i}^{2}}$$

Here $I_{i}$, $I_{r}$, $P_{i}$, and $P_{r}$ are the incident sound intensity, the reflected sound intensity, the incident sound pressure, and the reflected sound pressure, respectively. Supposing that $n=\frac{P_{\max}}{P_{\min}}=\frac{P_{i}+P_{r}}{P_{i}-P_{r}}$, Equation (17) can be converted to Equation (18).
(18)$$\alpha_{0}=1-\frac{P_{r}^{2}}{P_{i}^{2}}=\frac{4n}{{\left(n+1\right)}^{2}}$$

Acoustic pressure data in the standing wave can be achieved by the standing wave tube method. The difference in the sound level L between the peak value $L_{\max}$ and the valley value $L_{\min}$ can be calculated by Equation (19).
(19)$$L=L_{\max}-L_{\min}=20\mathrm{lg}P_{\max}/P_{0}-20\mathrm{lg}P_{\min}/P_{0}=20\mathrm{lg}P_{\max}/P_{\min}=20\mathrm{lg}n$$

According to Equations (17)–(19), the actual sound absorption coefficient $\alpha_{0}$ of MSC-AM with the vertical incidence can be achieved, as shown in Equation (20), and the peak value $L_{\max}$ and valley value $L_{\min}$ of the sound pressure for a certain frequency can be obtained by the standing wave tube tester automatically [45,46].
(20)$$\alpha_{0}=\frac{4\times 10^{\left(L_{\max}-L_{\min}\right)/20}}{{\left(1+10^{\left(L_{\max}-L_{\min}\right)/20}\right)}^{2}}$$

## 3. Results and Discussion

### 3.1. Theoretical Analysis Results

Distribution of the sound absorption coefficients of the MSC-AM in theory is shown in Figure 3. It can be found that the peak absorption frequency of MSC-AM is periodic. Within a frequency range of 200–2910 Hz, three frequency bands can be roughly divided according to the relationship between the length $L_{N}$ and wavelength λ of the spiral chamber and the distribution law of peak absorption frequency on the acoustic absorption coefficient curve, 200–1060 Hz, 1060–1950 Hz and 1950–2910 Hz, respectively. Sound-absorption frequency band I is concentrated near the frequency of the 1/4 wavelength of the spiral chamber in the MSC-AM, frequency band II is concentrated near the frequency of 1/2 wavelength and frequency band III is concentrated near the frequency of 3/4 wavelength, respectively.

For further analysis from the structural impedance, the impedance of MSC-AM is normalized, as shown in Equation (21).
(21)$$Z/Z_{0}=-iZ_{c}\cot(k_{c}L_{N})/Z_{0}\sigma =(Z_{c}/Z_{0})\left[-i\cot(k_{c}L_{N})/\sigma \right]$$

$Z_{0}$ is the characteristic impedance of air. To achieve perfect sound absorption, the impedance in the AM must match the characteristic impedance in the air, that is $Z/Z_{0}=1$. $Z_{c}/Z_{0}$ is close to 1.0 when the effective radius of the chamber section is large enough, such as at the millimeter level. In the case of a small perforation rate $\sigma_{N}$ (such as $\sigma_{N}<0.02$ in this research), $\cot(k_{c}L_{N})$ must be close to 0. The derivation can be $k_{c}L_{N}=\pi /2$, that is $L_{N}\approx \lambda /4$, where λ is the sound wavelength. When the length of the spiral chamber is about 1/4 of the wavelength of the incident sound, perfect sound absorption can be achieved. This principle well reveals spectral distribution of the theoretical sound absorption coefficient. At the same time, MSC-AM not only shows fine sound absorption in the 1/4 wavelength, but also has good sound absorption in the 1/2 wavelength and 3/4 wavelength ranges. It can be observed that a single spiral chamber in the MSC-AM structure not only produces a single absorption peak, but also produces the extra multiple groups of absorption peaks in each frequency band, which theoretically proves that the MSC-AM has potential excellent sound-absorption characteristics in the low-frequency and broadband sound absorption. Meanwhile, it can be found from Figure 3 that for a single spiral chamber, the peak sound-absorption coefficient is decreasing and the absorption bandwidth is increasing with the increase in frequency band. Moreover, it can be observed that the distance between the neighboring sound-absorption peaks is gradually increasing with the increase in the frequency band. Thus, generally speaking, sound-absorption frequency band I, concentrated near the frequency of the 1/4 wavelength of the spiral chamber, can obtain a better sound-absorption performance.

### 3.2. Finite Element Simulation Results

The finite element numerical simulation method is proposed to simulate the excitation response of the MSC-AM under the action of the simulated physical field to verify the effectiveness of the structure [47,48]. The utilized parameters of finite element simulation model in this research are as follows: size of the largest unit, 2 mm; size of the smallest unit, 0.02 mm; maximum growth rate of neighboring unit, 1.3; curvature factor, 0.2; resolution ratio of the narrow area, 1; mesh type, free tetrahedron mesh; layer number of the boundary area, 8; stretching factor of the boundary layer, 1.2; regulatory factor of the thickness of the boundary layer, 1. Based on the finite element simulation model, distribution of sound-absorption coefficients in the MSC-AM in simulation is obtained, as shown in Figure 4. According to the results of the theoretical analysis in Section 3.1, distribution of the sound-absorption coefficients in the MSC-AM in the simulation is also divided into three frequency bands within the same frequency range. The results in theory and those in simulation have a certain consistency in the distribution of absorption peaks, and also show the acoustic characteristics of multi-peak absorption.

According to the numerical simulation results, spiral chambers with different lengths are screened out and the sound pressure level (SPL) diagrams of the corresponding frequencies of the six spiral chambers at the absorption peak of MSC-AM under the action of simulated physical sound field are summarized, as shown in Figure 5. The response states of chambers with the same length under the excitation of an external sound field are the same, so six different spiral channels can represent the six groups of spiral chambers in the whole MSC-AM. The distribution of sound energy at the resonance frequency can be vividly reflected by the SPL diagram.

It can be observed from the SPL diagrams corresponding to the six absorption peaks in the frequency band I that the selected six spiral chambers are located at the different spiral radii and have different lengths. The spiral chamber near the center of the spiral has a shorter length and a higher resonance frequency, while the spiral chamber with a large spiral radius has a longer chamber length and a lower resonance frequency. The peak position of sound absorption is related to the length of the FP channel. The larger the screw radius is, the longer the chamber length is, and the peak absorption position of the chamber moves to the low-frequency direction. At the resonance frequency, the sound energy is highly concentrated in the chamber with a certain length, followed by the other surrounding chambers. The sound energy dissipation is chiefly caused by the resonance spiral chamber, while the other chambers play a small role in it.

Compared the frequency band I, the distribution of the peak absorption frequency band of II and III basically follows the rule of frequency band I, but there are obvious differences. First of all, from the SPL distribution, it can be observed that the sound energy density in frequency band I is highly concentrated in one group of the six spiral chambers, while the other chambers are relatively weak. At the peak absorption frequency of frequency band II and III, the concentration of the sound energy density is not strong, and this trend is further strengthened as it moves to the high-frequency regions. Secondly, in low-frequency band I, the interval between the sound absorption peaks is narrow, while the interval between the peak sound-absorption frequencies in the middle and high-frequency areas is significantly wide, which can be judged from Figure 5. In addition, the number of absorption peaks decreases, and the one-to-one correspondence with the number of chambers cannot be realized. The major reason may be that the peak absorption frequency interval increases as the peak absorption frequency moves to the middle and high-frequency regions, and there is partial overlap between frequency bands II and III for the absorption peaks of some chambers. Based on the analysis of the above reasons, it can be concluded that the acoustic characteristics of multi-peak absorption of the MSC-AM are relatively significant in the low-frequency region. As it moves to the high-frequency range, the peak absorption coefficient decreases and the peak absorption frequency interval increases, which limits the application of this structure in the high-frequency region.

In order to better reveal the sound absorption mechanism in the MSC-AM and explore the phase delay effect of the spiral chamber on the acoustic wave and the multiple-chambers coupling effect, the sound absorption characteristics of spiral chambers with different lengths are analyzed separately in frequency band I. In addition, the sound field and boundary conditions of the single spiral chamber simulation model and MSC-AM simulation model are identical. The sound absorption curve of a single spiral chamber and that of the MSC-AM are shown in Figure 6. It can be seen from Figure 6 that the peak absorption frequency of a single spiral chamber is located in the process of phase delay, jumping from 0 to *π*, which is consistent with the theoretical analysis. The peak frequency of the sound absorption of each single spiral chamber is in good agreement with that of the multi-chamber combination unit, but the sound absorption amplitude of the whole MSC-AM structure is significantly higher than that of a single chamber. In the low-frequency region, the peak frequency interval of sound absorption is relatively narrow, and the absorption coefficient between two absorption peaks of MSC-AM is much higher than that of the single chamber. In the higher-frequency region, the peak sound absorption frequency interval expands, and the sound absorption coefficient between two absorption peaks of MSC-AM decreased. It indicates that the sound absorption in the combination unit is the coupling effect of different chambers instead of the simple superposition of each single chamber. After the spiral chambers of different lengths are combined, the resonance frequency of a single spiral chamber does not change significantly, and the sound absorption peak of the entire MSC-AM is generated by the contribution of each spiral chamber. Furthermore, the coupling effect of multiple chambers significantly improves the sound absorption performance of the combined structure compared to a single chamber, showing an excellent broadband sound-absorption effect.

### 3.3. Experimental Results

Limited by the measurement range of the utilized AWA6290T standing wave tube tester [49,50,51], the experimental samples are tested in a frequency range of 200–1600 Hz. Comparisons of the sound absorption coefficients in the MSC-AM with those in theory and those in simulation are shown in Figure 7. In a bandwidth range Δ*f*_1_ = 320 Hz of 360–680 Hz, the actual sound absorption coefficients are larger than 0.8, exhibiting a distinguished sound-absorption performance. Meanwhile, the sound-absorption coefficient is above 0.5 in a range of Δ*f*_2_ = 1250 Hz of 350–1600 Hz. The major reason for excellent sound-absorption performance in the 360–680 Hz range is that the interval between the first absorption peaks of different spiral chambers is small in this range. Moreover, the absorption peaks of multiple chambers can better make up the valleys between the absorption peaks after the coupling and superposition. In a frequency range of 680–1600 Hz, the absorption peaks of the different spiral chambers expand, and the coupling effect among different chambers is not obvious. In the frequency band between neighboring absorption peaks, the corresponding sound absorption performance is weak relative to that at the resonance frequency region.

Taking the experimental data as a reference, it can be observed from Figure 7 that deviation in the theoretical data is larger than that of the simulation data. The major reason for this phenomenon is that there are many approximations and simplifications in the constructed theoretical model, and the finite element model can simulate the actual acoustic response accurately. Meanwhile, it can be seen that most of the resonance frequency points obtained in simulation are mainly consistent with the actual ones. In turn, it can be found that there seems to more resonance frequency points derived from the theoretical model relative to the experimental data. Moreover, sound-absorption coefficients at the resonance frequency points obtained in simulation are larger than the actual ones generally. This is mainly because there exists some manufacturing error for spiral chambers of the fabricated sample. In addition, the surface of the fabricated sample is not absolutely smooth relative to that in theory or that in simulation, which will lead to an additional increase in the actual sound energy viscous losses. Thus, for the sound-absorption valley frequencies, the results of actual sound-absorption coefficients are significantly higher than those in theory or those in simulation.

Structural parameters in the MSC-AM can be flexibly adjusted to meet the various requirements in sound-absorption performance and bandwidth. Meanwhile, the method to coil the chamber to the spiral can provide references for the design or improvement of other kinds of AM. Taking into account that the effective sound absorption of the FP resonator strictly depends on its dimension close to the 1/4 wavelength of the incident sound wave, the proposed MSC-AM can achieve a broadband sound absorption in the low-frequency range and small occupied space simultaneously, which exhibits significant potential applications in low-frequency noise reduction and suppression.

## 4. Conclusions

On the basis of the FP resonance principle, a novel sound absorber in the MSC-AM is proposed and studied in this research. Through the theoretical modeling, finite element simulation, sample preparation and experimental validation, the effectiveness and practicability of the MSC-AM are certified. The major achievements obtained in this research are as follows:

(1)The FP channel is curled to the spiral chamber through the ingenious structural design, which can significantly reduce the occupied space of the MSC-AM absorber and effectively realize a broadband sound absorption in the low-frequency range. In this study, six series of spiral chambers, with sequential lengths of 95.99 mm, 138.72 mm, 152.27 mm, 191.19 mm, 201.23 mm and 247.25 mm, are compactly arranged in a cylinder with a diameter of 100 mm and thickness of 80 mm, which form the proposed MSC-AM.(2)A theoretical model for sound-absorption performance of the MSC-AM is built according to the derivation of its acoustic impedance, which preliminarily proves that the MSC-AM can achieve broadband sound absorption in the low-frequency range. Moreover, the sound-absorption mechanism of MSC-AM is revealed through the finite element simulation, which indicates that under stimulation of the external sound field, perfect absorption is realized by the strong resonance within a certain frequency for each group of FP spiral chambers with the same length in the MSC-AM. Furthermore, parallel connections of the multiple spiral chambers with various lengths form the MSC-AM can obtain the broad bandwidth through the coupling of multiple absorption peaks.(3)The experimental sample of MSC-AM is prepared by an LFS 3D printer based on additive manufacturing, and its actual sound-absorption coefficients are measured by the AWA6290T standing wave tube tester. The experimental results indicate that the actual sound-absorption coefficients of the MSC-AM exceed 0.8 with a bandwidth of Δ*f*_1_ = 320 Hz in the 360–680 Hz range, which certifies that it can obtain excellent sound-absorption performance in the low-frequency range. Meanwhile, in the bandwidth of Δ*f*_2_ = 1250 Hz in the 350–1600 Hz range, its sound absorption coefficients are larger than 0.5, which proves that it can achieve broadband sound absorption in the low–middle frequency region.

## Figures and Tables

**Figure 1 materials-15-03882-f001:**
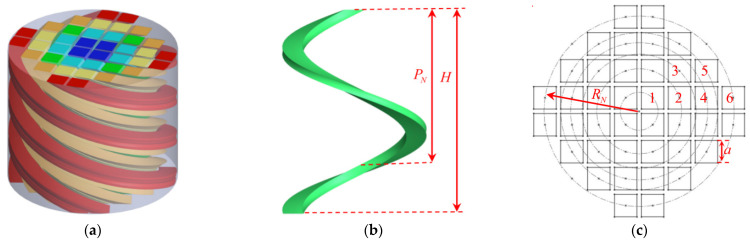
Schematic diagram of the MSC-AM structure. (**a**) 3D model of the structure; (**b**) 3D model of a single spiral chamber; (**c**) 2D sketch of the structure section.

**Figure 2 materials-15-03882-f002:**
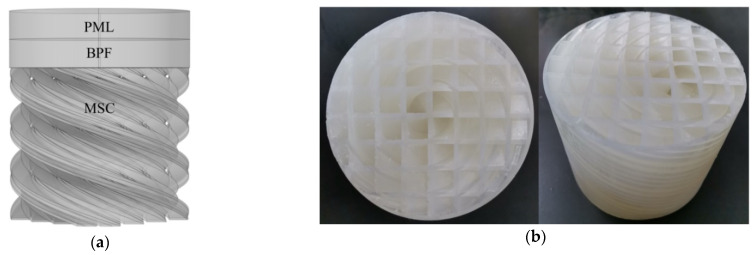
Finite element simulation analysis and experimental validation of sound absorption performance of the MSC-AM. (**a**) Finite element simulation model of the MSC-AM; (**b**) the prepared experimental sample for the MSC-AM; (**c**) schematic diagram of the AWA6290T standing wave tube tester.

**Figure 3 materials-15-03882-f003:**
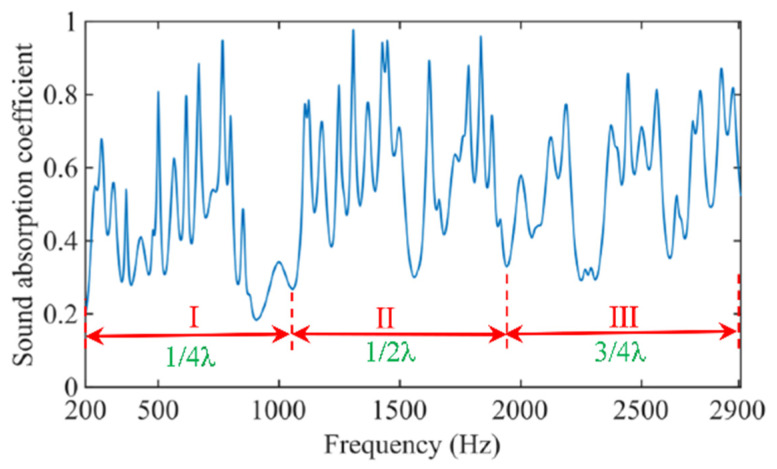
Distribution of the sound absorption coefficients of the MSC-AM in theory.

**Figure 4 materials-15-03882-f004:**
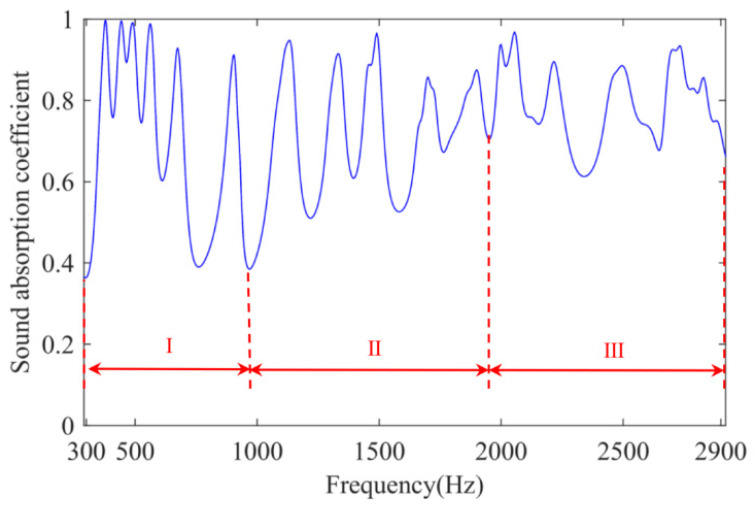
Distribution of the sound absorption coefficients in the MSC-AM in simulation.

**Figure 5 materials-15-03882-f005:**
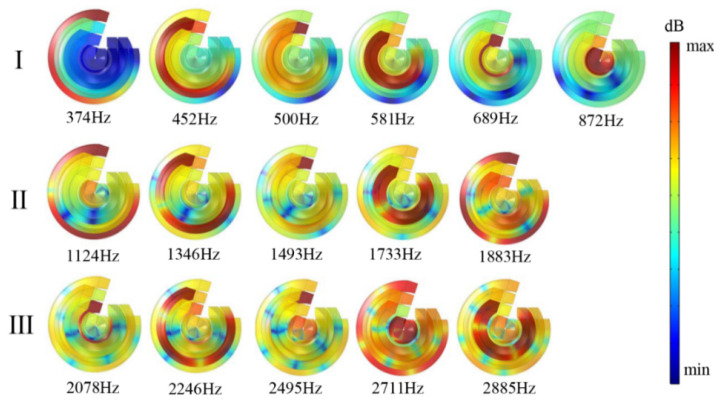
Distributions of the sound pressure level (SPL) of MSC-AM within resonance frequency points in simulation.

**Figure 6 materials-15-03882-f006:**
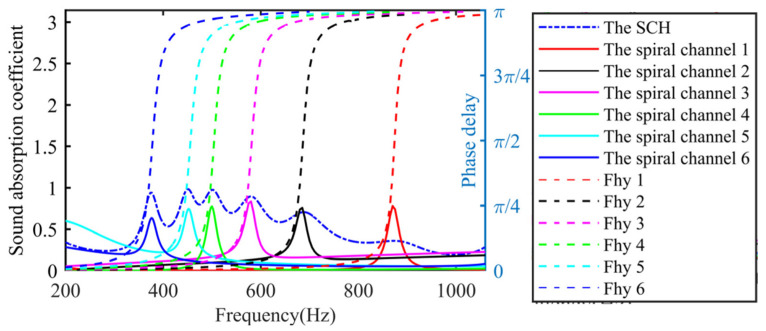
Contrast of the sound absorption coefficient and the phase delay in the MSC-AM, in which the dashed lines represent the curves of phase delay of the single spiral chamber, the solid lines represent the curves of sound absorption coefficient of the single spiral chamber, and the dot-dashed lines represent the curves of total sound absorption of the MSC-AM, respectively.

**Figure 7 materials-15-03882-f007:**
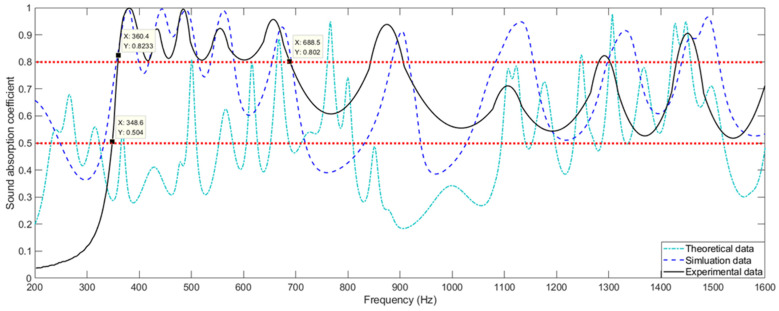
Comparisons of the actual sound absorption coefficients of the MSC-AM with those in theory and those in simulation.

**Table 1 materials-15-03882-t001:** Summary of the utilized constants in the theoretical model for the MSC-AM.

Parameters	Symbol	Unit	Values
Acoustic velocity of the air	$c_{0}$	m/s	343
Density of the air	$\rho_{0}$	${\mathrm{kg}/m}^{3}$	1.21
Standard atmospheric pressure	$P_{0}$	Pa	1.01325 × 10^5^
dynamic viscosity coefficient	μ	Pa⋅s	1.8 × 10^−5^
thermal conductivity	κ	$W/\left(m\cdot K\right)$	0.0258
specific heat capacity at the constant volume	$C_{v}$	$J/\left(\mathrm{kg}\cdot K\right)$	718
specific heat ratio of the air	γ	-	1.4

## Data Availability

The data that support the findings of this study are available from the corresponding author upon reasonable request.

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
