# Peer review of "Acoustic Metamaterials for Low-Frequency Noise Reduction Based on Parallel Connection of Multiple Spiral Chambers"

_materials, 2022, doi:10.3390/ma15113882_

Round 1
Reviewer 1 Report
Ref. Review: Materials 1726338
Paper Title: Acoustic Metamaterials for Low-Frequency Noise Reduction based on Parallel Connection of Multiple Spiral Chambers
Dear Authors and Editor,
Based on the text exposed in the entitled paper: "Acoustic Metamaterials for Low-Frequency Noise Reduction based on Parallel Connection of Multiple Spiral Chambers", I recommend minor revisions before acceptance and publishing on Materials.
This study investigates a novel sound absorber of the acoustic metamaterial by parallel connecting the 19 multiple spiral chambers (abbreviated as MSC-AM).
The idea looks interesting and could help with the absorbing materials technologies that require small spaces due to thin walls, which is an ordinary reality in the new buildings.
Please consider the following suggestions for improvement and further consideration for publication on Materials:
Introduction:
- Please reinforce in the text the problem of new buildings with small spaces with thin walls requiring new absorbing technologies.
- Theoretical Analysis and Experimental Validation:
- Figure 2: This diagram, with the design of the measurement chain, probably was based on other works. Could you add the references e which part is your adaptation? Also, inform the references in the initial text of section 3.
3.1 Finite Element Simulation Analysis:
- Please add references in the text exposed in both paragraphs.
3.2 Sample Preparation by Additive Manufacturing Technology:
- Please add references in the exposed text.
3.3. Experimental Verification by Standing Wave Tube:
- Lines 261-271: Could you show a figure of the waves’ behavior inside the tube? The illustration can help the reader to understand the text from section 3.3.
Results:
4.1. Theoretical Analysis Results:
- Please indicate in Figure 3 the findings noted in the above text (lines 297-310). Also, refer to the indications of frequency bands I, II and III from figure 3 in the text, highlighting which frequency bands the absorption worked well.
4.2. Finite Element Simulation Results:
- Lines 313-320: Here, you can also refer to where the multi-peak absorption worked better. In which frequency range?
Conclusions:
- Could you please indicate the limitations of the work?
Author Response
Response to reviewer 1
Thank you very much for your kind review to our manuscript and positive assessment to our research. We have revised the manuscript carefully according to your and other reviewers’ comments. The responses to your comments are as follows.
- Please reinforce in the text the problem of new buildings with small spaces with thin walls requiring new absorbing technologies.
Response:
Thank you very much for your kind suggestion. The related discussion on the problem of new buildings with small spaces with thin walls requiring new absorbing technologies is added in paragraph 1 of section “1. Introduction” to the revised manuscript. Meanwhile, the added part is highlighted in yellow.
- Figure 2: This diagram, with the design of the measurement chain, probably was based on other works. Could you add the references e which part is your adaptation? Also, inform the references in the initial text of section 3.
Response:
Thank you very much for your kind suggestion. The design of the measurement chain in the Figure 2(c) “Schematic diagram of the AWA6290T standing wave tube tester” is constructed based on the instrument's working principle and physical drawing, and it is supported by the Hangzhou Aihua Instruments Co., Ltd., Zhejiang, China, as mentioned in the presentation, which aimed to make the readers to better understand the detection process. Thus, except the tested sample, there is no additional adaptation to the instrument. Two references were added in the text according to your suggestion, both in the section 2.3 and section 3, and the added parts are highlighted in yellow.
- 3.1 Finite Element Simulation Analysis: Please add references in the text exposed in both paragraphs.
Response:
Thank you very much for your kind suggestion. We have added the references cited in the relevant content in section “Finite Element Simulation Analysis” in the revised manuscript, and the added parts are highlighted in yellow.
- 3.2 Sample Preparation by Additive Manufacturing Technology: Please add references in the exposed text.
Response:
Thank you very much for your kind suggestion. We have added the references cited in the relevant content in the revised manuscript, and the added part is highlighted in yellow.
- Lines 261-271: Could you show a figure of the waves’ behavior inside the tube? The illustration can help the reader to understand the text from section 3.3.
Response:
Thank you very much for your kind suggestion. The content of Figure 2(c) corresponds to the text of Section 3.3. In order to facilitate readers' understanding, we add the waves’ behavior inside the tube to Figure 2(c).
- Please indicate in Figure 3 the findings noted in the above text (lines 297-310). Also, refer to the indications of frequency bands I, II and III from figure 3 in the text, highlighting which frequency bands the absorption worked well.
Response:
Thank you very much for your kind suggestion. The proposed structure of the article currently has no mature theory for reference. We have spent a lot of work consulting the literature in related fields in an attempt to establish an accurate theoretical model. By the time the revised manuscript is returned, no better method is found to establish a theoretical model of the structure. However, the sound absorption coefficient prediction theoretical model derived by the acoustoelectric analogy method can preliminarily predict the potential value of the structure in the field of broadband noise control. The related presentations are added in the revised manuscript, and these modifications are highlighted in yellow.
- Lines 313-320: Here, you can also refer to where the multi-peak absorption worked better. In which frequency range?
Response:
Thank you very much for your kind inquiry. It can be found from Figure 3 that for a single spiral chamber, the peak sound absorption coefficient is decreasing and the absorption bandwidth is increasing with the increase of frequency band. Moreover, it can be observed that distance between the neighboring sound absorption peaks is larger gradually with the increase of frequency band. Thus, generally speaking, the sound absorption frequency band I concentrated near the frequency of the 1/4 wavelength of the spiral chamber can obtain a better sound absorption performance. The related presentations are added in the revised manuscript, and these modifications are highlighted in yellow.
- Could you please indicate the limitations of the work?
Response:
Thank you very much for your kind inquiry. We suppose the future direction to improve our work is to attempt to simplify the structure, because the spiral chamber is difficult to fabricate. Although additive manufacturing technology can manufacture the complex metastructure, its present production efficiency need further improvement for the mass production. Therefore, we will try to simplify the metamaterial, which aim to promote its practical application.

Reviewer 2 Report
- The low force stereolithography (LFS) 3D printing method which is used to prepare the experimental sample shouldbe explained in details.
- The methods, materials and settings used to make the sample(LFS) should be given in Section 2.
- It is recommended that section 3. “Theoretical Analysis and Experimental Validation” Merge in section 2. “Materials and Design”.
- Figure 1 (a) and 1 (c) are contradictory. Additional explanations should be provided.
- The color contrast in Figure 6 is low, making it difficult to detect.
- In the explanations of Figure 7, more explanations should be given for the difference between the actual and the theory and simulation.
- It is suggested that the conclusion be rewritten. The current text is too lengthy and parts of it should be moved to Section 4. “Results and Discussions”.
Author Response
Response to reviewer 2
Thank you very much for your kind review to our manuscript and positive assessment to our research. We have revised the manuscript carefully according to your and other reviewers’ comments. The responses to your comments are as follows.
- The low force stereolithography (LFS) 3D printing method which is used to prepare the experimental sample should be explained in details.
Response:
Thank you very much for your kind suggestion. We have modified the presentation about the low force stereolithography (LFS) 3D printing, and these modifications are highlighted in yellow in the revised manuscript.
- The methods, materials and settings used to make the sample (LFS) should be given in Section 2.
Response:
Thank you very much for your kind suggestion. We have added some parameters about the low force stereolithography (LFS) 3D printing in preparing the experimental sample, and these modifications are highlighted in yellow in the revised manuscript.
- It is recommended that section 3. “Theoretical Analysis and Experimental Validation” Merge in section 2. “Materials and Design”.
Response:
Thank you very much for your kind suggestion. Seriously considering the reviewers’ opinions, we merge section 3. “Theoretical Analysis and Experimental Validation” into section 2. “Materials and Design”. The revised part of the manuscript has been marked as yellow in the paper.
- Figure 1 (a) and 1 (c) are contradictory (矛盾). Additional explanations should be provided.
Response:
Thank you very much for your kind suggestion. In the Figure 1c, the marked number represent a group of chambers with the same parameters. Such as the number 1 represents the four blue chambers in the Figure 1a; number 3 represents the four green chambers in the Figure 1a; number 4 represents the eight yellow chambers in the Figure 1a; number 6 represents the eight red chambers in the Figure 1a. These added presentations are highlighted in yellow in the revised manuscript.
- The color contrast in Figure 6 is low, making it difficult to detect.
Response:
Thank you very much for your advice. In the revised manuscript, image resolution and line width are improved to make it easier to detect.
- In the explanations of Figure 7, more explanations should be given for the difference between the actual and the theory and simulation.
Response:
Thank you very much for your kind suggestion. Seriously considering the reviewers’ opinions, the proposed structure of the article currently has no mature theory for reference. We have spent a lot of work consulting the literature in related fields in an attempt to establish an accurate theoretical model. By the time the revised manuscript is returned, no better method is found to establish a theoretical model of the structure. However, the sound absorption coefficient prediction theoretical model derived by the acoustoelectric analogy method can preliminarily predict the potential value of the structure in the field of broadband noise control. This part of the work is meaningful. In addition, in order to make the research results more accurate and convincing, the sound absorption characteristics of the structure are simulated and verified by experiments. It is easy to see from the comparison of sound absorption coefficient curves between simulation values and experimental values that their distribution laws are in good agreement, and both show the effect of broadband sound absorption. The causes of their errors are also analyzed in the manuscript. The revised part of the manuscript has been marked as yellow in the paper.
- It is suggested that the conclusion be rewritten. The current text is too lengthy and parts of it should be moved to Section 4. “Results and Discussions”.
Response:
Thank you very much for your kind suggestion. We have revised the conclusion section carefully according to your and other reviewers’ comments, and these modifications are highlighted in yellow in the revised manuscript.

Reviewer 3 Report
Dear authors, a manuscript ‘Acoustic Metamaterials for Low-Frequency Noise Reduction based on Parallel Connection of Multiple Spiral Chambers’, Manuscript ID: materials-1726338, have some weakness that must be improved significantly.
Please find some of the most crucial suggestions, as listed below:
- The ‘Introduction’ section is interesting and contains plenty of useful knowledge review, nevertheless, describing the noise, various frequencies (bandwidth) and its human being influence should be, even mentioned, without limitations to the selected frequencies only. Please find below some examples of a manuscript that considered different bandwidths of the noise:
- https://doi.org/10.1016/j.cirp.2014.03.086
- https://doi.org/10.3390/ma14175096
- https://doi.org/10.1017/jfm.2013.477
- According to the above, first comment, there is no detailed critical review of the topic in its current form the ‘Introduction’ section is more with a description of the current knowledge than presenting some lacks that should be fulfilled (by the results in the manuscript presented). Please try to improve your literature review with some critical statements that would indicate the author(s) motivation of the studies provided.
- The references were selected appropriately many of them (more/less 22) are from the last 5-6 years (2017 and newer). However, the form of the referenced item is ver poor, e.g. in ref.1 the surnames and names were mistaken (replaced) and surnames were abbreviated. This section must be unified as well.
- There is no word, e.g. in section 2.2. (similar to the section 3.3.), if all of the equations were newly proposed by the author(s) or, respectively, presented in a previous paper(s). From that matter, the novelty in equations must be highlighted, if exist.
- Still, according to the ‘Theoretical modelling’ (2.2.) section, the selection of some of the variables, further used for calculations, was not justified and, consequently, looks like chosen arbitrarily. Please try to provide some clarifications (!).
- There are many shortcuts and abbreviations in the text that findings are difficult to follow for a regular reader and, simultaneously, make him confused. Please provide an additional section, e.g. Abbreviations, to make some improvement in text reading and, respectively, validation of the study's significance.
- According to the sentence ‘The sound absorption frequency band I is concentrated near the frequency of the 1/4 wavelength of the spiral chamber in the MSC-AM, the frequency band II is concentrated near the frequency of 1/2 wavelength, and the frequency band III is concentrated near the frequency of 3/4 wavelength respectively.’, lines 289-293, the axes of ‘concentrations’ should be indicated in Figure 3, e.g. by the arrows, or other, suitable way.
- Figure 7 must be re-worked so that, even easy to follow, in some cases (e.g. for the first peak for experimental data) it is difficult to find, even approximately, the Frequency value along the horizontal axis. This figure is crucial and its presentation must be much more confidential.
- From the ‘Conclusion’ section, except for many detailed data, e.g. values of variables, there is missing some (one) general conclusion, taking the whole novelty into account and presenting against the current state of knowledge.
- Improving the above (previous) suggestion, please try to make shorter the conclusions or, simultaneously, divide them into more separate and numbered gaps that in their current form is too long and, unfortunately, similar to some sentences presented in the manuscript body text. Even repeated, sentences should be re-written (in other words) in the ‘Conclusion’ section, if allowed.
Generally, the proposed manuscript has some weaknesses and, at least in the current form, is not suitable for publication in a quality journal as the Materials is.
The manuscript must be improved significantly (!) before any further processing, if allowed.
Author Response
Response to reviewer 3
Thank you very much for your kind review to our manuscript and positive assessment to our research. We have revised the manuscript carefully according to your and other reviewers’ comments. The responses to your comments are as follows.
- The ‘Introduction’ section is interesting and contains plenty of useful knowledge review, nevertheless, describing the noise, various frequencies (bandwidth) and its human being influence should be, even mentioned, without limitations to the selected frequencies only. Please find below some examples of a manuscript that considered different bandwidths of the noise:
https://doi.org/10.1016/j.cirp.2014.03.086
https://doi.org/10.3390/ma14175096
https://doi.org/10.1017/jfm.2013.477
Response:
Thank you very much for your kind suggestion. The suggested references are added in the introduction section, and the corresponding modifications are highlighted in yellow.
- According to the above, first comment, there is no detailed critical review of the topic in its current form the ‘Introduction’ section is more with a description of the current knowledge than presenting some lacks that should be fulfilled (by the results in the manuscript presented). Please try to improve your literature review with some critical statements that would indicate the author(s) motivation of the studies provided.
Response:
Thank you very much for your kind suggestion. The introduction mainly summarizes the current research status and lists the research results that have attracted more attention. Then put forward the common deficiencies of current research work: in order to achieve the coupling of different sound absorption frequencies, the different chamber combinations with different structural parameters are required, which results in the difficulty of compact arrangement of the combined structures and waste of space. According to the suggestions of you and other reviewers, we reviewed relevant researches in the introduction and expounded the differences and innovations of this work. Relevant content has been marked in yellow in the revised manuscript.
- The references were selected appropriately many of them (more/less 22) are from the last 5-6 years (2017 and newer). However, the form of the referenced item is ver poor, e.g. in ref.1 the surnames and names were mistaken (replaced) and surnames were abbreviated. This section must be unified as well.
Response:
Thank you very much for your kind suggestion. We have carefully checked and unified all the reference format in the article, and corrected some of the reference format errors in the revised manuscript, so that it can meet the requirements of the journal.
- There is no word, e.g. in section 2.2. (similar to the section 3.3.), if all of the equations were newly proposed by the author(s) or, respectively, presented in a previous paper(s). From that matter, the novelty in equations must be highlighted, if exist.
Response: Thank you very much for your kind suggestion.
The equations in Section 2.2 is derived on the basis of previous research results, and has been distinguished in the revised draft. The equations in Section 3.3 is mainly the current classical Standing Wave Tube test principle, and references are added to the relevant equations in the revised manuscript to make the paper more rigorous.
- Still, according to the ‘Theoretical modelling’ (2.2.) section, the selection of some of the variables, further used for calculations, was not justified and, consequently, looks like chosen arbitrarily. Please try to provide some clarifications (!).
Response:
Thank you very much for your kind suggestion. As we mentioned at the end paragraph of the section “2.2. Theoretical modeling”, Structural parameters of the proposed MSC-AM can be adjusted to achieve the variable sound absorption performance for the different requirements. Thus, we use a group parameters to verify the feasibility of the proposed MSC-AM, and there is no special selection of the parameters. These structural parameters of the MSC-AM can be adjusted to realize the customization of sound absorption spectrum for various applications.
- There are many shortcuts and abbreviations in the text that findings are difficult to follow for a regular reader and, simultaneously, make him confused. Please provide an additional section, e.g. Abbreviations, to make some improvement in text reading and, respectively, validation of the study's significance.
Response:
Thank you very much for your kind comment. In order to improve readability and conciseness of the paper, the abbreviations of repeated phrases have been processed, and corresponding notes have been given where the abbreviations first appear. Of course, if necessary, additional content can be provided in the form of supplementary material for the paper.
- According to the sentence ‘The sound absorption frequency band I is concentrated near the frequency of the 1/4 wavelength of the spiral chamber in the MSC-AM, the frequency band II is concentrated near the frequency of 1/2 wavelength, and the frequency band III is concentrated near the frequency of 3/4 wavelength respectively.’, lines 289-293, the axes of ‘concentrations’ should be indicated in Figure 3, e.g. by the arrows, or other, suitable way.
Response:
Thank you very much for your kind comment. We have modified the Figure 3 according to your suggestion.
- Figure 7 must be re-worked so that, even easy to follow, in some cases (e.g. for the first peak for experimental data) it is difficult to find, even approximately, the Frequency value along the horizontal axis. This figure is crucial and its presentation must be much more confidential.
Response:
Thank you very much for your kind comment. We have modified the Figure 7 according to your suggestion.
- From the ‘Conclusion’ section, except for many detailed data, e.g. values of variables, there is missing some (one) general conclusion, taking the whole novelty into account and presenting against the current state of knowledge.
Response:
Thank you very much for your kind suggestion. We have revised the conclusion section carefully according to your and other reviewers’ comments.
- Improving the above (previous) suggestion, please try to make shorter the conclusions or, simultaneously, divide them into more separate and numbered gaps that in their current form is too long and, unfortunately, similar to some sentences presented in the manuscript body text. Even repeated, sentences should be re-written (in other words) in the ‘Conclusion’ section, if allowed.
Response:
Thank you very much for your kind suggestion. We have revised the conclusion section carefully according to your and other reviewers’ comments.
- Generally, the proposed manuscript has some weaknesses and, at least in the current form, is not suitable for publication in a quality journal as the Materials is. The manuscript must be improved significantly (!) before any further processing, if allowed.
Response:
We tried our best to improve the manuscript and made some changes in the manuscript according to the suggestions of all reviewers. We appreciate for Editors/Reviewers’ warm work, and hope that the correction will meet with approval. We are looking forward to the result of final acceptance.

Reviewer 4 Report
Authors reported the development of a sound absorber of the acoustic metamaterial following Fabry-Perot resonance principle. The numerical and experimental results are interesting and this paper can be accepted for publication after addressing the following minor comments:
- Add finite element modeling framework in detail, i.e., strong and weak form of governing equations, element discretization, solution, etc.
- There are few typos and grammar errors.
- In the recent past, topology optimization has been a tool to design acoustic metamaterials with wideband width e.g., https://doi.org/10.1016/j.compstruct.2022.115389 https://doi.org/10.1016/j.compstruct.2021.114846 . Discuss this aspect also in the introduction section.
- Reduce the length of conclusion section and add key points only.
Author Response
Response to reviewer 4
Thank you very much for your kind review to our manuscript and positive assessment to our research. We have revised the manuscript carefully according to your and other reviewers’ comments. The responses to your comments are as follows.
- Add finite element modeling framework in detail, i.e., strong and weak form of governing equations, element discretization, solution, etc.
Response:
Thank you very much for your kind suggestion. The finite element simulation part of this paper is realized through the acoustic module of COMSOL software, and the parameters and boundary settings of the simulation model are introduced in the section “3.2. Finite Element Simulation Results”. The software is proved to be effective in the field of thermal viscous acoustic simulation. Due to the limitation of paper length, the detailed introduction of simulation model is weakened. According to your kind suggestion, some essential parameters for the finite element model are added in the revised manuscript, and they are highlighted in yellow.
- There are few typos and grammar errors.
Response:
Thank you very much for your kind suggestion. The English in the manuscript is corrected and revised carefully, and the existed grammar and spelling mistakes are modified. Meanwhile, these modifications are highlighted in yellow in the revised manuscript.
- In the recent past, topology optimization has been a tool to design acoustic metamaterials with wideband width e.g., https://doi.org/10.1016/j.compstruct.2022.115389 https://doi.org/10.1016/j.compstruct.2021.114846 . Discuss this aspect also in the introduction section.
Response:
Thanks a lot for your kind suggestion. More updated literatures were added in the introduction section in the revised manuscript according to your and other reviewers’ comments, and the further reviews of understanding of the optimization method of broadband sound absorption structure were cited, which aimed to improve feasibility demonstration of this study and get more complete trajectory of the field. Meanwhile, these modifications are highlighted in yellow in the revised manuscript.
- Reduce the length of conclusion section and add key points only.
Response:
Thank you very much for your kind suggestion. We have revised the conclusion section carefully according to your and other reviewers’ comments.

Round 2
Reviewer 2 Report
As authors have performed an adequate revise, the manuscript might be accepted for publication.
Author Response
Response to reviewer 2
Thank you very much for your kind review to our manuscript and positive assessment to our research. We have revised the manuscript carefully according to your and other reviewers’ comments. The responses to your comments are as follows.
- As authors have performed an adequate revise, the manuscript might be accepted for publication.
Response:
Thank you very much for your kind review to our manuscript and positive assessment to our research. We have further revised the manuscript according to the editor’s and other reviewers’ comments, which aims to make the manuscript more readable and reasonable. Thank you again for your kind review and support.

Reviewer 3 Report
Dear authors, a manuscript titled ‘Acoustic Metamaterials for Low-Frequency Noise Reduction based on Parallel Connection of Multiple Spiral Chambers’, Manuscript ID: materials-1726338, have been improved in a required manner, therefore can be further processed by the editorials of the Materials journal.
However, before final decision, please let me ask you for two, of the last issues:
1. I feel that the ‘Conclusions’ section is too weak, and has some general (flat) responses. This section does not correspond to the whole manuscript, which is much better than the results presented in the conclusions. Therefore, please try to emphasize novelty more strongly. Moreover, conclusions should be divided into separate, numbered gaps.
2. The DOI numbers, as far as I know, and the Materials template requirements, should be presented in the ‘References’.
Taking the above, two suggestions, the paper can be accepted for publication in a quality journal as the Materials is.
In the end, thank you for your responses that, in their current form, were addressed properly and make the manuscript suitable for publication in the Materials journal, as mentioned previously.
Author Response
Response to reviewer 3
Thank you very much for your kind review to our manuscript and positive assessment to our research. We have revised the manuscript carefully according to your and other reviewers’ comments. The responses to your comments are as follows.
- Dear authors, a manuscript titled ‘Acoustic Metamaterials for Low-Frequency Noise Reduction based on Parallel Connection of Multiple Spiral Chambers’, Manuscript ID: materials-1726338, have been improved in a required manner, therefore can be further processed by the editorials of the Materials journal. However, before final decision, please let me ask you for two, of the last issues.
Response:
Thank you very much for your kind review to our manuscript and positive assessment to our research. We have further revised the manuscript according to your comments carefully.
- I feel that the ‘Conclusions’ section is too weak, and has some general (flat) responses. This section does not correspond to the whole manuscript, which is much better than the results presented in the conclusions. Therefore, please try to emphasize novelty more strongly. Moreover, conclusions should be divided into separate, numbered gaps.
Response:
Thank you very much for your kind suggestion. We have adjusted the ‘Conclusions’ section and the ‘Results and Discussions’ section. Meanwhile, the presentations are further modified to make the manuscript more readable and reasonable.
- The DOI numbers, as far as I know, and the Materials template requirements, should be presented in the ‘References’.
Response:
Thank you very much for your kind suggestion. The DOI numbers of all the references are added in the ‘References’ of the revised manuscript.
- Taking the above, two suggestions, the paper can be accepted for publication in a quality journal as the Materials is. In the end, thank you for your responses that, in their current form, were addressed properly and make the manuscript suitable for publication in the Materials journal, as mentioned previously.
Response:
Thank you very much for your kind review to our manuscript and positive assessment to our research. We have further revised the manuscript according to your and the other reviewers’ comments, which aims to make the manuscript more readable and reasonable. Thank you again for your kind review and support.
